# Effects of High-Intensity Interval Training on Selected Adipokines and Cardiometabolic Risk Markers in Normal-Weight and Overweight/Obese Young Males—A Pre-Post Test Trial

**DOI:** 10.3390/biology11060853

**Published:** 2022-06-02

**Authors:** Nejmeddine Ouerghi, Mohamed Kacem Ben Fradj, Martine Duclos, Anissa Bouassida, Moncef Feki, Katja Weiss, Beat Knechtle

**Affiliations:** 1High Institute of Sport and Physical Education of Kef, University of Jendouba, UR13JS01, Jendouba 7100, Tunisia; najm_ouerghi@hotmail.com (N.O.); bouassida_anissa@yahoo.fr (A.B.); 2Rabta Hospital, Faculty of Medicine of Tunis, University of Tunis El Manar, LR99ES11, Tunis 1007, Tunisia; kacim1984@hotmail.fr (M.K.B.F.); monssef.feki@gmail.com (M.F.); 3Departments of Sport Medicine and Functional Explorations, University-Hospital, G. Montpied Hospital, Clermont-Ferrand, F-63003 Clermont-Ferrand, France; mduclos@chu-clermontferrand.fr; 4INRA, UNH, CRNH Auvergne, F-63000 Clermont-Ferrand, France; 5UFR Medicine, Clermont University, University of Auvergne, BP 10448, F-63000 Clermont-Ferrand, France; 6Medbase St. Gallen Am Vadianplatz, 9000 St. Gallen, Switzerland; katja@weiss.co.com; 7Institute of Primary Care, University of Zurich, 8001 Zurich, Switzerland

**Keywords:** adipokine, cardiometabolic health, insulin resistance, intermittent training, obesity

## Abstract

**Simple Summary:**

Adipose tissue secretes bioactive substances called adipokines that affect cardiometabolic health. High-intensity interval training has a beneficial effect on cardiometabolic health, but its impact on adipokines is unclear. This research aims to assess the effects of 8-week high-intensity interval training on plasma levels of four adipokines (leptin, adiponectin, chemerin, and omentin-1) and cardiometabolic risk markers in normal- and excess-weight young males. The findings suggest a beneficial role of adiponectin and omentin, and a harmful role of leptin and chemerin in cardiometabolic health. Following the training, plasma omentin-1 levels had increased in normal- and excess-weight youth, but adiponectin, leptin, and chemerin levels had not changed in both groups. Body mass, fat mass, plasma lipids, and insulin sensitivity improved in excess-weight group, only. Body composition seems not to influence the response of adipokines to high-intensity interval training.

**Abstract:**

The study aimed to assess effects of high-intensity interval training (HIIT) on plasma adipokines and cardiometabolic markers in normal and excess weight youth. Eighteen healthy young males (18.2 ± 1.06 yrs.) were divided in normal-weight group (NWG; body mass index (BMI), 20.5 ± 1.51 kg/m^2^; n = 9) and excess-weight group (EWG; BMI, 30.8 ± 4.56 kg/m^2^; n = 9). Participants performed an eight-week HIIT program without caloric restriction. Body composition, plasma leptin, adiponectin, chemerin, omentin-1, lipids, C-reactive protein (CRP), and the homeostasis model assessment index for insulin resistance (HOMA-IR) were assessed before and after the HIIT program. The program resulted in significant increases in omentin levels (*p* < 0.01) in EWG (27%) and NWG (22%), but no changes in leptin, adiponectin, and chemerin in both groups. BMI (−1.62%; *p* = 0.015), body fat (−1.59%; *p* = 0.021), total cholesterol (−11.8%; *p* = 0.026), triglycerides (−21.3%; *p* = 0.023), and HOMA-IR (−31.5%; *p* = 0.043) decreased in EWG only. Repeated measures detected significant interaction “Time x Group” for body mass and BMI only. Eight-week HIIT program improved body composition, lipid profile, and insulin sensitivity in excess-weight individuals. It resulted in an increase in omentin levels in both normal- and excess-weight groups, but no changes in leptin, adiponectin, and chemerin. Body composition has not influenced the response of the four adipokines to HIIT.

## 1. Introduction

Adipose tissue is an active endocrine organ that secretes a variety of bioactive molecules, called adipokines [1]. These compounds signal to targets in a variety of tissues and organs, including liver, skeletal muscle, brain, immune cells, and adipose tissue itself, modulating energy balance, glucose and lipid metabolisms, vascular, neuromuscular, immune functions, and inflammatory response [1]. Adipokines impact cardiometabolic health by exerting either beneficial or harmful effects. Leptin is recognized as proinflammatory and deleterious, while adiponectin is considered as anti-inflammatory and beneficial for cardiometabolic health [1,2,3,4]. Impacts of other adipokines such as omentin and chemerin are unresolved, with data suggesting both beneficial and harmful roles [5,6,7,8,9]. Adipokines are tightly connected with body composition, insulin resistance, and inflammation, which make them strongly involved in obesity and obesity-associated chronic diseases [1,2,10]. Therefore, modulation of adipokine secretion and signaling, i.e., reduction of offensive adipokines and hyper secretion of defensive adipokines, may be relevant for the prevention and treatment of these deleterious conditions.

Regular physical activity is an efficient strategy for improving cardiometabolic health. It prevents obesity and visceral fat accumulation and reduces the risk of metabolic and cardiovascular diseases [11]. High intensity interval training (HIIT) is a mode of training that alternates high-intensity periods of working out and periods of rest, which was proven to induce weight loss and to improve cardiometabolic health [11,12]. Research on effects of HIIT on adipose tissue-derived adipokines has yielded conflicting results. There is evidence that this mode of training affects adipose tissue adipokine expression and release [13,14,15,16,17,18,19,20,21,22,23,24,25,26,27,28]. However, several studies showed no such effect [12,29,30,31,32,33,34,35,36].

Most previous studies that examined effect of HIIT on adipokines mainly focused on leptin and/or adiponectin, often showing a decrease in leptin and/or an increase in adiponectin plasma concentrations [13,15,16,17,18,20,21,22,23,25,26,28,29,30,31,33]. Few studies examined omentin’s response to HIIT, most of which showed an increase in the adipokine levels [14,27]. The scarce research on chemerin response to HIIT is inconclusive [24]. Almost all previous studies involved middle-aged, overweight/obese, or diseased individuals at high cardiometabolic risk. The effect of HIIT on adipokines in younger subjects at low cardiometabolic risk has been rarely investigated [23,31]. Moreover, whether potential changes in adipokines are due to changes in body composition/adiposity or to direct or indirect effects on muscle itself is unclear. Simultaneous measurement of adipokine changes with cardiometabolic markers changes may further elucidate the topic.

Previous studies generally focused on one adipokine, often involved subjects at high cardiometabolic risk, and did not firmly clear up the issue. In this context, we undertook this research to examine effect of HIIT on four adipose tissue-derived adipokines (i.e., leptin, adiponectin, chemerin, and omentin-1) and key cardiometabolic risk markers in young people at low cardiometabolic risk. To test how body composition affects adipokines’ response, we examined adipokines’ responses in normal-weight and overweight/obese individuals separately. To clarify the mechanisms underlying HIIT effects, we assessed associations between changes in adipokines and changes in markers of adiposity, circulating lipids, insulin resistance, and inflammation. We hypothesized that HIIT alters adipokines’ profile, likely due to an improvement in body composition and cardiometabolic status.

## 2. Materials and Methods

### 2.1. Ethical Approval

The study was conducted in accordance with the declaration of Helsinki and was approved by the local ethics committee (Rabta Hospital, HRT-15/18). All participants gave their written consent.

### 2.2. Participants

A total of 55 students from two pre-final secondary education classes in Dahmani (Tunisia) were asked to participate in the study. Eligible to participate in the study were students of both sexes, aged 17 to 20 years, who provided personal or parent/guardian written informed consent. Students with acute or chronic illness, those under medication, regular smokers or alcohol drinkers, and trained athletes were not included. Students who withdrew their consent, were absent for more than two sessions of training program, or missed pre- or post-program blood collection were excluded from analysis. An a priori power analysis was conducted using G*Power software [37], with an assumed type I error of 0.05 and a type II error rate of 0.20 (80% statistical power). It revealed that 16 participants in total would be sufficient to observe medium “Time × Group” interaction effects (f = 0.40). A larger sample was recruited to take account of possible dropouts. No girl wished to participate in the study. Thirty-three male students consented to participate to the study. Based on self-reported individual weight and height, twenty students were randomly selected after stratification on body mass index (BMI) into a normal-weight group (NWG; BMI < 25 kg/m^2^; n = 10) and an excess-weight group (EWG; BMI ≥ 25 kg/m^2^; n = 10). While participating in the training program, two students missed several sessions and were excluded from the study. Eighteen students, nine from the NWG and nine from the EWG, completed the entire program. The usual physical activity of these participants consisted of two-hour physical education lessons, weekly.

### 2.3. Study Protocol

The study was conducted from February to April 2014. During this period, the training program took place, and the assessments were carried out under the following environmental conditions; temperature varying between 17 °C and 23 °C and humidity varying from 70% to 75%. Body composition, cardiorespiratory fitness indices, and biochemical parameter measurements were performed in all participants before and after the training program. Body height and body mass were obtained, with subjects barefooted and lightly clothed, using a wall-mounted stadiometer and a calibrated scale. BMI (kg/m^2^) was calculated as body mass (kg) divided by the square of height (m). Skinfold thickness was determined in triplicate at four sites (biceps, triceps, subscapular, and suprailiac), using a calibrated Harpenden calliper (Holtain Instruments, Pembrokeshire, UK). The mean of the three values was recorded for each site. Body density (D) was calculated according to the equations of Durnin and Wormersley [38] for men aged 17–19 years as D = 1.162 − [0.063 (log Σ)] and for men aged 20–29 years as D = 1.1631 − [0.0632 (log Σ)], where Σ is the sum of the four skinfolds (in mm). Percentage body fat (BF) was calculated from D using Siri’s equation [39] as follows: BF = (4.95/D − 4.50) × 100. Maximal aerobic velocity (MAV) and maximal oxygen uptake (VO_2max_) were determined by a graded exercise test until exhaustion [40]. Heart rate was recorded during the test using a Polar heart rate monitor (Polar S810, Kempele, Finland); the highest recorded value was considered as the maximal heart rate (HRmax). All participants were familiarized with the incremental test during the week preceding the training program [41].

### 2.4. Training Program

The training program was carried out as previously described [41,42] during 8 consecutive weeks with 3 sessions per week. Each training session consisted of a 15-min warm-up, main stage, and 10-min cool-down stage. Warm-up consisted of 10-min continuous jogging at moderate intensity; 50% of maximal aerobic velocity (MAV), followed by 5-min dynamic stretching exercises and 5 short bursts of 20-m accelerations. The main stage consisted of two series of 30-s run at 100–110% of MAV, interspersed by periods of active recovery of 30-s run at 50% of MAV. Training progression was carried out by increasing the number of repetitions from 8 to 10 repetitions from the third week and increasing the intensity of work from the fifth week (5% increase of the MAV every two weeks). Finally, participants were cooled down by running at low intensity and performing static stretching during 10 min. No diet restriction was applied, but the students were instructed to maintain their usual eating and behavioral habits during the training period.

### 2.5. Blood Sampling and Methods of Analysis

Fasting blood samples were collected in the morning (around 8 a.m.) the day before the start and two days after the completion of the training program. The blood samples were centrifuged at 2000× *g* for 25 min and the plasma was frozen at −40 °C until analysis (within 6 months). Plasma adipokines were assessed in duplicate using specific commercially ELISA kits according to the manufacturer’s protocols (R&D Systems, Inc. Minneapolis, MN for leptin and adiponectin; Biovendor-Laboratorni-medicinaa.s., Karasek, Czech Republic for chemerin and omentin-1). Plasma insulin concentration was measured by a chemiluminescence immunoassay using a Liaison analyzer (DiaSorin Inc., Stillwater, MN) and the respective reagents kit. Plasma glucose, total cholesterol (TC), triglycerides (TG), and C-reactive protein (CRP) were assessed on an Architect C8000 analyzer using respective reagents kits (Abbott Laboratories, Abbott Park, IL). The homeostasis model assessment (HOMA-IR) was used as an estimate for insulin resistance; HOMA − IR= [fasting insulin (μU/mL) × fasting glucose (mmol/L)/22.5] [43].

### 2.6. Statistical Analysis

The analysis was performed using the Statistical Package for the Social Sciences version 18.0 software (SPSS Inc., Chicago, IL, USA). Data of continuous variables were examined for normality using the Kolmogorov–Smirnov test. Training-induced changes (∆), in circulating adipokines, and cardiometabolic markers were calculated as the difference between post-training and pre-training values. Comparisons between groups were carried out using independent-samples *t*-test, and within each group (post-training vs. pre-training) using paired-samples *t*-test. Correlations between continuous variables were tested using the Pearson test. Two-way (Time × Group) repeated measures analyses of variance were applied to test the combined effect (interaction) of HIIT and body composition on dependent variables (i.e., adipokines and cardiometabolic variables). Partial eta-squared (η_p_^2^) was calculated as a measure of effect size. A two-sided *p*-value < 0.05 was considered significant.

## 3. Results

### 3.1. Associations of the Adipokines with Cardiometabolic Risk Markers

At baseline, EWG showed higher leptin and chemerin and lower adiponectin andomentin-1 plasma concentrations than NWG. BMI, BF, TC, TG, CRP, and HOMA-IR were significantly higher, whereas VO_2max_ was significantly lower (*p* < 0.01) in EWG (Table 1). In a combined group of participants (NWG and EWG), pre- and post-training leptin and chemerin were positively correlated, while adiponectin and omentin-1 were negatively correlated with cardiometabolic risk markers (Table 2).

### 3.2. Adipokines Responses to HIIT

The eight-week HIIT resulted in a significant increase on plasma omentin-1 levels in both EWG and NWG. However, no significant changes were found for leptin, adiponectin, and chemerin in the two groups (Table 1, Figure 1). HIIT resulted in a decrease in BMI (−1.62%; *p* = 0.015), BF (−1.59%; *p* = 0.021), TC (−11.8%; *p* = 0.026), TG (−21.3%; *p* = 0.023), and HOMA-IR (−31.5%; *p* = 0.043) in EWG, but no significant changes were observed in NWG. Following the HIIT program, VO_2max_ increased but CRP did not change in both groups. Repeated measurements detected significant interactions group × time for body mass and BMI. No interaction group × time were found for the four adipokines and other variables investigated (Table 1).

### 3.3. Correlations of Training-Induced Adipokines and Cardiometabolic Markers Changes

HIIT-induced changes in chemerin were positively related to changes in BMI and TG. Adiponectin changes were negatively related to changes in TG and CRP. No other correlations were found between adipokines and cardiometabolic markers changes (Table 2).

## 4. Discussion

The study showed that leptin and chemerin are positively related, while adiponectin and omentin are negatively related to the cardiometabolic risk markers. HIIT resulted in an increase in plasma omentin-1 levels in both groups, but no significant changes in leptin, adiponectin, or chemerin levels were observed. Body composition, lipid profile, and insulin sensitivity improved in EWG only. CRP did not change in both groups. These findings corroborate the respective negative and positive association of leptin and adiponectin with cardiometabolic health [1,2,3,4] and suggest a detrimental role of chemerin and a protecting role of omentin for cardiometabolic health. The study findings suggest that body composition has no or little influence on adipokines’ response to HIIT since the responses are similar in normal weight and overweight/obese groups. Potential changes in adipokines appear not to be solely related to change in adiposity since leptin, adiponectin, and chemerin levels did not change despite weight loss and improved cardiometabolic status. Additionally, omentin increased in both groups, while adiposity changed only in the overweight/obese group.

In agreement with our finding, plasma omentin-1 has substantially increased in healthy and type 2 diabetes individuals [14], and in overweight/obese women [4,19] following HIIT programs. However, omentin did not change after 12 weeks of moderate interval training in obese males [44]. Response of omentin to aerobic/resistance training was also inconsistent, with studies showing either increased [45,46,47,48] or unchanged [49,50] levels. In this study and other studies [14,27,45,46,47,48], omentin increased in parallel with a reduction in body mass and an improvement in cardiometabolic status. Since omentin is lower in obese individuals [51] and since diet/lifestyle interventions-induced weight loss increased omentin levels [8,52], it could be assumed that omentin increases due to the training-induced weight loss. Nonetheless, omentin levels did not change after physical training, despite significant reductions in adiposity and insulin sensitivity [44,49,50]. In the present study, omentin has increased in parallel with a decrease in adiposity, plasma lipids, and insulin resistance in weight-excess individuals, but with no changes in these markers in normal-weight individuals. Thus, even if body composition and metabolic changes influence omentin response, the changes also occur in response to physiological adaptation of muscle to physical exercise. Omentin is preferentially produced by stromal vascular cells and monocytes/macrophages within omental and pericardial adipose tissue but seems not to be expressed within muscle cells [53]. It is likely that muscle-derived myokines released into circulation in response to physical exercise act on the adipose tissue, influencing omentin metabolism and secretion [46,53]. The present study showed that omentin changes independently of body composition, since it increases in both normal weight and overweigh/obese groups. Omentin’s increase with no change in adiposity in normal weight individuals suggests that the change is not only related to weight loss. Given its potential anti-inflammatory, antidiabetic, and antiatherogenic properties [53,54], omentin’s increase could be considered beneficial for health. Further research should be undertaken to shed light on the response of omentin to HIIT as well as the role of the adipokine in cardiometabolic health.

The study showed no changes in leptin and adiponectin, despite a significant decrease in adiposity, plasma lipids, and insulin resistance in weight-excess individuals. Literature reported inconsistent responses of the two adipokines to HIIT as well as to aerobic/resistance training. Following HIIT, plasma leptin decreased in parallel with [16,17,18,20,25,26] or without [15,21,23] weight loss in overweight/obese individuals. However, the levels remained unchanged with either reduction [12,28,30] or no change in adiposity in overweight/obese [29,32,33,34] and in normal weight [31] individuals. The response of adiponectin was also inconsistent; the levels increased in parallel with [13,18,20,25,26,27,28] or without [21,22] weight loss in overweight/obese individuals. However, adiponectin has remained unchanged despite weight loss [12,36] or without changes in body/fat mass among overweight/obese [33,34,35,55] and normal-weight [31] individuals. Finally, adiponectin has decreased with no significant changes in body mass and metabolic traits in overweight/obese subjects following short- or long-term HIIT programs [19,56,57]. A meta-analysis concluded that physical exercise resulted in a reduction in leptin and an increase in adiponectin plasma concentrations in obese young people [58]. Discrepancies may originate from differences in the populations investigated (e.g., body composition, physical fitness, dietary intake) and HIIT programs (e.g., duration, intensity, number of repetitions). However, changes more likely occur for longer programs. Most of the long-term HIIT programs (>12 weeks) resulted in decreased leptin and/or increased adiponectin levels, which are often associated with cardiometabolic status improvement [13,16,17,18,20,25,26,27,28].

Little research has been achieved regarding the effects of physical exercise on plasma chemerin. In disagreement with the study finding, two studies reported a decrease in chemerin following 6 weeks of HIIT [24] or endurance training [59] programs. Despite showing no change in chemerin, this study revealed positive associations with changes in cardiometabolic risk markers such as BMI and TG. Such a finding suggests that that chemerin is more affected by changes in adiposity and lipid metabolism than by physical training. While a precise role of chemerin in cardiometabolic health is unclear with evidence linking the adipokine to increased [9,60] or decreased [5,61] risk, our study findings rather support a detrimental cardiometabolic role of chemerin.

The present study is original by analyzing the response of four adipokines to HIIT, alongside changes in key markers of cardiometabolic risk, separately in normal-weight and overweight/obese subjects. Splitting participants according to body composition permitted us to conclude that adiposity has no or little influence on adipokine response to HIIT. The study has some limitations that should be stated. Due to the small sample size, the study could have missed significant changes in leptin, adiponectin, and chemerin. The training program may not be long enough to produce significant changes in these adipokines. It is likely that HIIT-induced changes in adiposity were too small to affect adipokine levels. Body fat was estimated based on the skinfold thickness method, which is not as accurate as DEXA scan or MR imaging for the purpose. Lack of a control group does not allow attributing observed changes to the sole effect of the HIIT program with high confidence. The study did not control for dietary intake and energy expenditure. Moreover, participants’ usual physical activity was not objectively measured using pedometers or assessed using questionnaire. However, no participant changed their eating habits or routine comportment during the training program period. This makes it unlikely that these factors influenced the results. Finally, the study involved mildly active young males, which makes the findings unsuitable for other categories of individuals. Given the above-mentioned methodological limitations, the study results should be considered with caution.

## 5. Conclusions

The study results suggest beneficial roles of adiponectin and omentin, and harmful roles of leptin and chemerin in cardiometabolic health. An eight-week HIIT program without calorie restriction was effective in decreasing adiposity, plasma lipids, and insulin resistance. A concomitant increase in omentin could contribute to this beneficial effect. However, the study failed to set up significant changes in leptin, adiponectin, and chemerin, probably due to lack of statistical power, short training program duration, and limited change in adiposity. Large trials applying uniform intervention protocol and longer training programs are needed to shed light on the effects of HIIT on adipokines production and uncover the underlying mechanisms.

## Figures and Tables

**Figure 1 biology-11-00853-f001:**
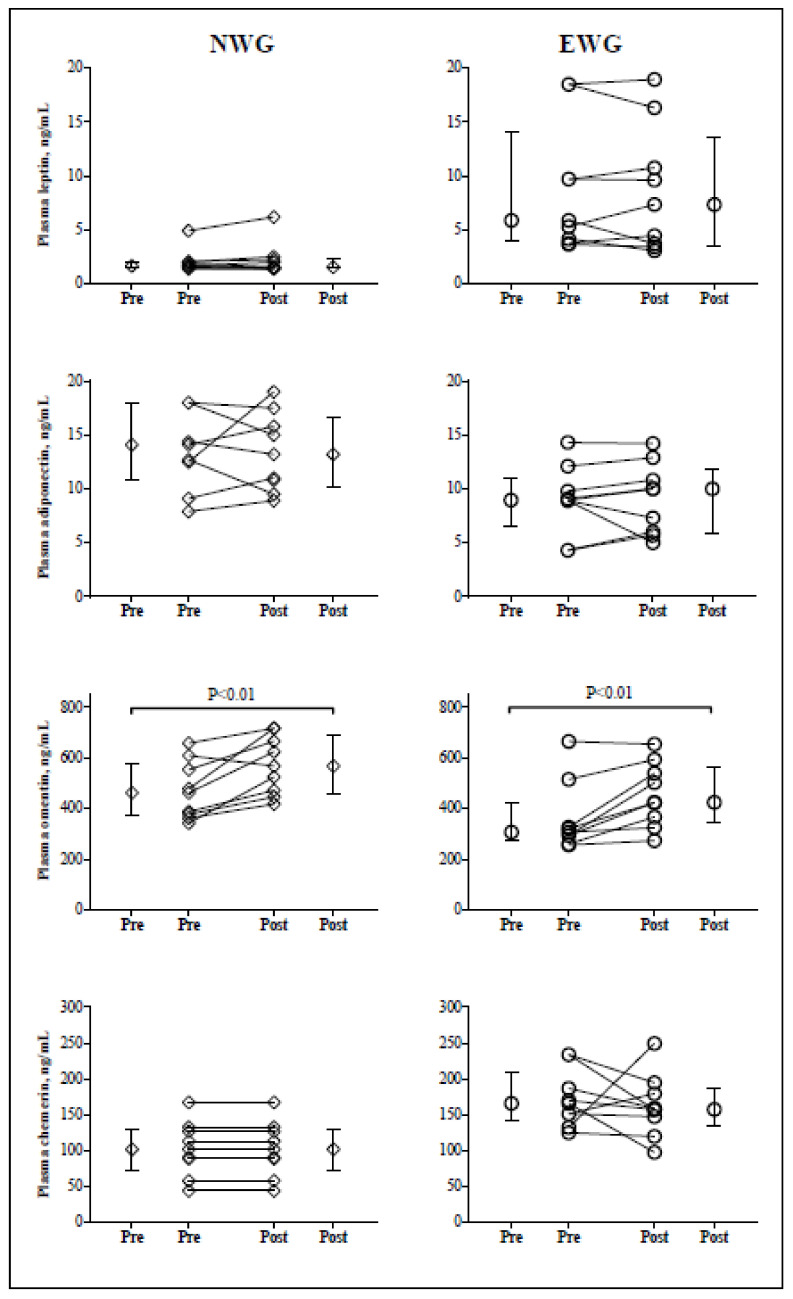
Changes in circulating Omentin-1, adiponectin, leptin, and chemerin concentrations following eight-week high-intensity interval training (HIIT) among normal-weight group (NWG) and excess-weight group (EWG); P-value (compared to pre-training program in the same group).

**Table 1 biology-11-00853-t001:** Pre- and post-training values and interaction time × group for anthropometrical, physiological, and biochemical measures in normal-weight and excess-weight groups.

	Normal-Weight Group (n = 9)		Excess-Weight Group (n = 9)		Interaction (Time × Group) ^a^	
	Pre-HIIT	Post-HIIT	Pre-HIIT	Post-HIIT	F	η_p_^2^
Age, years	18.1 ± 0.93		18.3 ± 1.22			
Body mass, kg	62.6 ± 4.61	62.5 ± 4.90	93.7 ± 16.9 ^###^	92.0 ± 15.9 *	5.96 ^†^	0.271
Body mass index, kg/m^2^	20.5 ± 1.51	20.5 ± 1.67	30.8 ± 4.56 ^###^	30.3 ± 4.25 *	6.24 ^†^	0.280
Body fat, %	12.0 ± 3.28	11.9 ± 3.10	22.5 ± 1.87 ^###^	22.1 ± 1.82 *	2.17	0.119
MAV, km/h	14.9 ± 0.53	15.4 ± 0.74 **	11.5 ± 1.15 ^###^	12.1 ± 0.96 **	0.64	0.039
VO_2max_, mL/kg/min	54.1 ± 1.84	55.6 ± 2.58 **	42.0 ± 4.03 ^###^	44.2 ± 3.37 **	0.58	0.035
Heart rate max, beat/min	190 ± 10.0	189 ± 10.2	193 ± 8.65	192 ± 7.58	0.03	0.002
Total cholesterol, mg/dL	136 ± 20.0	127 ± 19.8	171 ± 36.2 ^###^	150 ± 15.0 *	2.39	0.130
Triglycerides, mg/dL	81.6 ± 30.9	68.4 ± 16.1	122 ± 39.3 ^###^	89.7 ± 21.4 *	1.25	0.072
HOMA-IR	1.87 ± 1.43	1.26 ± 0.51	4.99 ± 2.62 ^##^	3.12 ± 1.47 *	1.96	0.108
C-reactive protein, mg/L	0.85 ± 0.67	1.27 ± 1.42	2.78 ± 1.57 ^##^	2.97 ± 1.77	0.98	0.006
Leptin, ng/mL	2.05 ± 1.09	2.23 ± 1.54	8.78 ± 5.96 ^##^	8.62 ± 5.84	0.47	0.028
Chemerin, ng/mL	103 ± 37.9	106 ± 41.9	172 ± 39.6 ^##^	162 ± 43.5	0.38	0.023
Adiponectin, ng/mL	14.2 ± 4.23	13.4 ± 3.61	8.96 ± 3.21 ^##^	9.10 ± 3.28	0.30	0.018
Omentin-1, ng/mL	470 ± 114	572 ± 115 **	359 ± 138 ^#^	455 ± 126 **	0.27	0.020

Data are expressed as mean ± SD; HOMA-IR, homoeostasis model assessment index for insulin resistance; MAV, maximal aerobic velocity; VO_2max_, maximal oxygen uptake; TC, total cholesterol; TG, triglycerides; *, *p* < 0.05; **, *p* < 0.01 (comparison in each group was achieved by paired-samples *t*-test with pre-training value as reference); ^#^, *p* < 0.05; ^##^, *p* < 0.01; ^###^, *p* < 0.001 (comparison between groups was achieved by independent-samples *t*-test with normal-weight group as reference). ^a^, interaction was tested using two-way repeated measures ANOVA (^†^, *p* < 0.05).

**Table 2 biology-11-00853-t002:** Correlations of pre-training (Pre) and post-training (Post) adipokines levels with cardiometabolic risk markers and of their respective training-induced changes.

		BMI	BF	TC	TG	CRP	HOMA-IR
Leptin	Pre Post	0.874 *** 0.848 ***	0.687 ** 0.673 ***	0.702 *** 0.651 **	0.805 *** 0.634 ***	0.767 *** 0.599 **	0.619 ** 0.860 ***
Chemerin	Pre Post	0.782 *** 0.573 *	0.767 *** 0.684 **	0.687 ** 0.244	0.775 *** 0.562 *	0.766 *** 0.591 **	0.674 ** 0.555 *
Adiponectin	Pre Post	−0.582 * −0.547 *	−0.448 −0.563 *	−0.379 −0.454	−0.590 −0.480 *	−0.564 * −0.681 **	−0.369 −0.415
Omentin-1	Pre Post	−0.552 * −0.547 *	−0.585 * −0.563 *	−0.531 * −0.369	−0.385 −0.523*	−0.411 −0.263	−0.409 −0.268
		∆ BMI	∆ BF	∆ TC	∆ TG	∆ CRP	∆ HOMA-IR
∆ Leptin		−0.345	−0.331	0.389	0.386	0.038	0.037
∆ Chemerin		0.133	-0.203	0.271	0.585 **	0.269	−0.010
∆ Adiponectin		−0.247	0.209	−0.196	−0.545 *	−0.561 *	−0.192
∆ Omentin-1		0.088	0.294	−0.094	−0.424	0.025	0.231

Values are Pearson correlation coefficients; BF, body fat; BMI, body mass index; CRP, C-reactive protein; HOMA-IR, homoeostasis model assessment index for insulin resistance; TC, total cholesterol; TG, triglycerides; ∆ = post-training value—pre-training value; *, *p* < 0.05; **, *p* < 0.01; ***, *p* < 0.001 (Pearson correlation).

## Data Availability

The datasets used and analyzed during the current study are available from the corresponding author on reasonable request.

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
