# Peer review of "Effects of High-Intensity Interval Training on Selected Adipokines and Cardiometabolic Risk Markers in Normal-Weight and Overweight/Obese Young Males—A Pre-Post Test Trial"

_biology, 2022, doi:10.3390/biology11060853_

Round 1

Reviewer 1 Report

Thank you for your replies!

Author Response

Reviewer 1

Thank you for your replies!

Response: We thank the reviewer for her/his good appreciation of the manuscript and help to improve its quality.

Reviewer 2 Report

Thanks for giving me this opportunity to review this paper. The present study aimed to assess the effects of high-intensity interval training (HIIT) on plasma adipokines and cardiometabolic markers in normal and excess weight youth. A total of 18 healthy young males performed an eight-week HIIT program. Before and after the interventions, blood samples were collected to measure different adipokines and cardiometabolic markers. The authors concluded that the eight-week HIIT program improved body composition, lipid profile, and insulin sensitivity in excess-weight individuals. It resulted in an increase in omentin levels in both normal- and excess-weight groups, but no changes in leptin, adiponectin, and chemerin. Body composition has not influenced the response of the four adipokines to HIIT. The research topic is interesting, and the strength of the present study was the complicated measurements of different adipokines and cardiometabolic markers. However, I have several concerns that may be considered by the authors to further improve their paper. See the below comments for the detail.

  1. The title should use “young males” rather than “youth” as all participants are males. Also, it is not a common presentation in using “excess-weight”, which may be changed to “overweight/obese”.
  2. The background information is not enough and the rationale of the present study is not clear. Therefore, the introduction should be revised substantially. E.g., why is it necessary to investigate these adipokines and cardiometabolic markers? What is the exact research gap? What are the expected contributions to the literature? These important questions were not well explained in the introduction part.
  3. Methods, participants, may provide clear inclusion and exclusion criteria.
  4. Page 3, line 113, reference 18 is not correct, please update.
  5. Statistic, page 4, line 151-152, why use Spearman rank test?
  6. Discussion, should highlight and discuss the key findings first.
  7. In the discussions, the authors only listed the previous findings. For me, I think that it is not clear what are the similarities and differences between the present study and the previous studies for each adipokine and cardiometabolic risk marker. Further discussions are needed to highlight the new findings of the present study.
  8. Table 3 is not necessary for an original study. It should be included in a review paper.
  9. Several obvious limitations have been mentioned in the discussion, which is good. However, because of these limitations, the explanations of the results and the conclusions should be very cautious.

Author Response

Reviewer 2

The research topic is interesting, and the strength of the present study was the complicated measurements of different adipokines and cardiometabolic markers. However, I have several concerns that may be considered by the authors to further improve their paper. See the below comments for the detail.

Response. We thank the reviewer for her/his analysis of the manuscript . We hope that all comments/questions have been adequately addressed below.

  1. The title should use “young males” rather than “youth” as all participants are males. Also, it is not a common presentation in using “excess-weight”, which may be changed to “overweight/obese”.

Response. The title was changed as follows “Effects of high-intensity interval training on selected adipokines and cardiometabolic risk markers in normal-weight and excess-weight youths overweight/obese young males. A pre-post test trial”

  1. The background information is not enough and the rationale of the present study is not clear. Therefore, the introduction should be revised substantially. E.g., why is it necessary to investigate these adipokines and cardiometabolic markers? What is the exact research gap? What are the expected contributions to the literature? These important questions were not well explained in the introduction part.

Response. The introduction has been substantially modified to address the issues raised by the reviewer (please see the revised introduction section).

  1. Methods, participants, may provide clear inclusion and exclusion criteria.

Response. Clear inclusion and exclusion criteria were provided as follow “A total of 55 students from two pre-final secondary education classes in Dahmani (Tunisia) were asked to participate in the study. Were eligible to participate in the study students of both sexes, aged 17 to 20 years who provided personal or parent/guardian written informed consent. Students with acute or chronic illness, those under medication, regular smokers or alcohol drinkers, and trained athletes were not included. Students who withdraw their consent, are absent during more than two sessions of training program or miss pre- or post-program blood collection are excluded from analysis”.

  1. Page 3, line 113, reference 18 is not correct, please update.

Response. Many thanks to the reviewer for pointing out this error. We have corrected this error in the revised. MAV and VO2max were determined according to Cazorla's paper (reference 40 in the revised list). The position of the references has changed after the modifications we made to the introduction section according to suggestions of the two reviewers

  1. Statistic, page 4, line 151-152, why use Spearman rank test?

Response. We thank the reviewer for pointing this issue. Since adipokines and cardiometabolic markers variables are normally distributed, Pearson test is perfect for testing the correlation between these variables. In the revised version, we changed “Spearman rank” by “Pearson” in statistical analysis section (line 197) and in table 2 legend (line 238). In table 2, we reported the values of Pearson coefficients instead of Spearman coefficients. The values have changed but the associations and signification have remained unchanged.

  1. Discussion, should highlight and discuss the key findings first.

Response. The main findings were highlighted and discussed first (please see the revised discussion section).

  1. In the discussions, the authors only listed the previous findings. For me, I think that it is not clear what are the similarities and differences between the present study and the previous studies for each adipokine and cardiometabolic risk marker. Further discussions are needed to highlight the new findings of the present study.

Response. The discussion has been revised taking into consideration the comments and suggestions of the two reviewers. We commented our findings on in the light of literature data.

Since the literature is widely conflicting on the effect of HITT and aerobic/resistance training on adipokines production, our results agree with some studies but disagree with others. It is true that this study has several limitations that are pointed out in discussion, and which necessitate caution when considering the findings .

  1. Table 3 is not necessary for an original study. It should be included in a review paper.

Response. Table 3 has been deleted.

  1. Several obvious limitations have been mentioned in the discussion, which is good. However, because of these limitations, the explanations of the results and the conclusions should be very cautious.

Response. The following sentences were added to the discussion section “Given the above-mentioned methodological limitations, the study results should be considered with caution” (lines 336-337), and “However, the study failed to set up significant changes in leptin, adiponectin and chemerin, probably due to lack of statistical power, short training program duration and limited change in adiposity.” (lines 363-365).

Reviewer 3 Report

Line 38: the word “groups” are missing.

Introduction: Please explain why adipokines are important in the context of obesity and its relationship with cardiometabolic health. Moreover, the state-of-art of adipokines regulation in response to different exercise types is missing.

How did you evaluate the caloric restriction ? How can you sure that the individuals had caloric restriction ?
In the literature, several studies have been reported an effect of high intensity exercise on hormones-regulating appetite, such as ghrelin, acylated ghrelin, glucagon-like peptide 1, and YY peptide. Interestingly, none of them was evaluated.

A dietary questionnaire was applied?

The cardiometabolic health-related parameters should be clarify in the methods section

Author Response

Reviewer 3

Response: We thank the reviewer for her/his analysis of the manuscript . We hope that all comments/questions have been adequately addressed below.

Line 38: the word “groups” are missing.

Response. Thank you for your remark. We believe the word "groups" is unnecessary in this sentence since "NWG" stands for "normal weight group" and "OWG" stands for "overweight/obese group".

Introduction: Please explain why adipokines are important in the context of obesity and its relationship with cardiometabolic health. Moreover, the state-of-art of adipokines regulation in response to different exercise types is missing.

Response. The introduction section was revised while explaining the importance of adipokines assessement in the context of obesity and its relationship with cardiometabolic health as follows “These compounds signals to targets in a variety of tissues and organs including liver, skeletal muscle, brain, immune cells, and adipose tissue itself, modulating energy balance, glucose and lipid metabolisms, vascular, neuromuscular, and immune functions, and inflammatory response [1]. Adipokines impact cardiometabolic health by exerting either beneficial or harmful effects” and “Adipokines are tightly connected with body composition, insulin sensitivity, and inflammation, which make them strongly involved in obesity and obesity-associated chronic diseases [1,2,10]. Therefore, modulation of adipokine secretion and signaling, i.e. reduction of offensive adipokines and hyper secretion of defensive adipokines, may be relevant for the prevention and treatment of these deleterious conditions”. (please see the introduction section).

We also described the state-of-art of adipokines regulation in response to interval training as follows “Research on effects of HIIT on adipose tissue-derived adipokines has yielded conflicting results. There is evidence that this mode of training affects adipose tissue adipokine expression and release [13-28]. However, several studies showed no such effect [12,29-36]. Most previous studies that examined effect of HIIT on adipokines mainly focused on leptin and/or adiponectin, often showing a decrease in leptin and/or an increase in adiponectin plasma concentrations [13,15-18,20-23,25,26,28-31,33]. Few studies examined omentin’s response to HIIT, most of which showed an increase in the adipokine levels [14,27]. The scarce research on chemerin response to HIIT is inconclusive [24]. (please see the introduction section).

How did you evaluate the caloric restriction? How can you sure that the individuals had caloric restriction?

Response. The study aimed to assess the effect of physical training (i.e., HIIT) on selected adipokines independently of dietary intervention. Thus, we instructed to the participants maintain their usual eating and behavioral habits during the training period, which allow to avoid potential confounding effect of diet on adipokine secretion. This information was provided in the text (lines 173-174)

In the literature, several studies have been reported an effect of high intensity exercise on hormones-regulating appetite, such as ghrelin, acylated ghrelin, glucagon-like peptide 1, and YY peptide. Interestingly, none of them was evaluated.

Response. We agree with the reviewer that investigating effect of high intensity exercise on hormones-regulating appetite, such as ghrelin, acylated ghrelin, glucagon-like peptide 1, and YY peptide is of great interest.  We have examined ghrelin response to HIIT in a previous experimental study and in a review of literature. However, the present study focused on the four adipokines. 

A dietary questionnaire was applied?

Response. No dietary questionnaire was applied. Participants were instructed to maintain their usual eating during the training period. However, this check was not objectively verified

The cardiometabolic health-related parameters should be clarify in the methods section

Response. In the Methods section, we described how we determined BMI and BF in the subsection “Study protocol”, and TC, HDL-C, TG, CRP and HOMA-IR in the subsection “Blood sampling and methods of analysis 

Round 2

Reviewer 2 Report

Thanks for the authors' efforts in addressing my concerns. I have no further comments.

This manuscript is a resubmission of an earlier submission. The following is a list of the peer review reports and author responses from that submission.

Round 1

Reviewer 1 Report

The authors aimed to assess the effects of high-intensity interval training on plasma adipokines and cardiometabolic markers in normal and excess weight youth.

This is an interesting manuscript with several strenght. 

I just have a few minor comments to make.

Keywords

I would delete the keyword "interval training" and replace it, if necessary, with another word other than those in the text. To optimize the search for the manuscript through search engines, it is always advisable to use keywords other than the title.

Materials and Methods

The methods are described clearly and correctly.

In the paragraph "Participants", however, I would insert the a priori power analysis done before the study to justify the sample size (n = 18). The sample size is quite small but as these are repeated measurements it can be fine.

For example: "An a priori power analysis (G*Power; Faul, Erdfelder, Lang, & Buchner, 2007) with an assumed type I error of 0.05 and a type II error rate of 0.20 (80% statistical power) was calculated and revealed that 16 participants in total would be sufficient to observe medium “Time x Group” interaction effects (f = 0.39). A larger sample was recruited to take account of possible drop-outs". 

Discussion

Table 3 should be inserted in the "Results" section. Remove the table from the Discussions where the results are interpreted and commented on.

The limitations are properly discussed.

Reviewer 2 Report

Dear Authors 

the paper entitled "Effects of high-intensity interval training on selected adipokines and cardiometabolic risk markers in normal-weight and excess-weight youths. A pre-post test trial" describes the impact of HIIT training on body composition, adipokine and cardiometabolic risk parameters.

The reviewer raises some critical issues to address:

  • the present study replicates some results presented by a previous studies carried out by the same authors (1.Ouerghi N, Fradj M, Bezrati I, et al. Effects of high-intensity interval training on body composition, aerobic and anaerobic performance and plasma lipids in overweight/obese and normal-weight young men. Biology of Sport. 2017;34(4):385-392. doi:10.5114/biolsport.2017.69827 2.Ouerghi N, Ben Fradj MK, Bezrati I, Feki M, Kaabachi N, Bouassida A. Effect of High-Intensity Interval Training on Plasma Omentin-1 Concentration in Overweight/Obese and Normal-Weight Youth. Obes Facts. 2017;10(4):323-331. doi: 10.1159/000471882).
  • the aim: "We also tested how potential changes in adipokines depend 
    on changes in adiposity, circulating lipids, insulin resistance and the inflammatory status. We hypothesized that HIIT alters adipokine profiles likely due to an improvement in cardiometabolic status. " Should be revise in a different point of view: the word depend should be revise with an approach such "the relationship between adipokines and ...."
  • Being a "random selection after stratification on body mass index (BMI)" procedure, how many eligible students were available for the study?
  • line 90: "The study was conducted from February to April" what was the year? Actually is April 11 2022.
  • "The temperature varied between 17°C and 23°C and the humidity ranged from 70% to 75%." the reviewer assumes that both the temperature and the environment in which the assessments are carried out and the environment in which the training program took place. Please specify better.
  • the reference 17 no describe a methodologic procedures of body composition. Please replace it with a specification by which you understand which sites are ranked and what conversion formula was used. In addition, is not clear the rational of this reference at line 102.
  • line 104: for better readability and understanding of the manuscript it would be advisable to describe the training program as well as refer to other studies.
  • the sentence at line 107: "The usual physical activity of these participants consisted of two-hour physical education lessons, weekly." it cannot be considered as an assessment of physical activity levels. It would be necessary to evaluate this activity also with questionnaires.
  • line 150 and 183: "Following the HIIT program, VO2max increased but CRP did not change in both groups." Table 1 report an increase of C-reactive protein, mg/L in NWG and in EWG.
  • fig. 1 appear not necessary, report the same information of table 1 and also these information are in the text.
  • the organization of table 2 is not clear, what do the authors want to prove? the relationship between the raw parameters or the relationship between the changes? choose which parameters to insert.
  • The rationale for evaluating the relationships between changes in adipokines and changes in body composition / cardiometabolic risk is unclear to the reviewer if no significant changes were recorded in leptin, chemerin, and adiponectin (Table 1).
  • table 3 not appear necessary for the aim of the study